CGRP overexpression does not alter depression-like behavior in mice

Hashikawa-Hobara Narumi hobara@dls.ous.ac.jp
Otsuka Ami
Okujima Chihiro
Hashikawa Naoya
Department of Life Science, Okayama University of Science , Okayama , Japan
van der Westhuizen Francois
Electronic publication date: 2021 Jul 2
Publication date: 2021
Volume: 9
Electronic Location ID: e11720
Received 2021 Apr 14; Accepted 2021 Jun 13
Copyright: © 2021 Hashikawa-Hobara et al.
Copyright year: 2021
Copyright holder: Hashikawa-Hobara et al.
License: This is an open access article distributed under the terms of the Creative Commons Attribution License, which permits unrestricted use, distribution, reproduction and adaptation in any medium and for any purpose provided that it is properly attributed. For attribution, the original author(s), title, publication source (PeerJ) and either DOI or URL of the article must be cited.
License URL: https://creativecommons.org/licenses/by/4.0/

Keywords: CGRP, Transgenic, Mice, Depressive-like behavior, BDNF, Akt/mTOR pathway

Funding: The authors received no funding for this work.

==============================
Background

The calcitonin gene-related peptide (CGRP) is a neuropeptide that is released from capsaicin-sensitive nerves as a potent vasodilator involved in nociceptive transmission. While CGRP has been rigorously studied for its role in migraines owing to its vasodilation and inflammation activities, the effects of CGRP overexpression on depressive-like behaviors remain insufficiently understood.

Methods

In the present study, we performed a battery of behavioral tests, including the social interaction test, open field test, and sucrose preference test, to evaluate social defeat stress using male C57BL6J or CGRP overexpression in transgenic (Tg) mice (CGRP Tg). We performed mRNA and protein analyses on the brain-derived neurotrophic factor (BDNF), phosphorylated Akt, mTOR, and p70S6K in the hippocampi.

Results

CGRP Tg mice showed increased levels of Bdnf mRNAs, low locomotor activity, and no deficits in social interaction, which indicate that CGRP Tg mice exhibit stress resistance and not depression. However, the open field test significantly decreased after 15-day social defeat stress exposure. We also evaluated depressive-like behavior using the sucrose preference and social interaction tests. Our data indicate that defeated CGRP Tg mice exhibited a depressive-like phenotype, which was inferred from increased social avoidance and reduced sucrose preference compared with non-defeated controls. Although stress exposure did not change the BDNF levels in CGRP Tg mice, it significantly decreased the expression levels of p-Akt, p-mTOR and p-p70S6K in the mice hippocampi. We conclude that CGRP-overexpressing Tg mice have normal sensitivity to stress and down-regulated hippocampal Akt/mTOR/p70S6K pathways.

Introduction

Depression is a life-threatening disorder with high rates of suicide. Most patients with depression suffer from social, occupational, educational, and interpersonal relationship problems. Symptoms of depression range from emotional and cognitive impairments to systemic dysfunctions. Several studies have demonstrated that the mechanisms underlying the psychopathology of depression are multifaceted. Major depressive disorder leads to a smaller hippocampus (Sheline et al., 1999), and decreased or impaired neurogenesis in the hippocampus has been reported (Joëls et al., 2004; Berton & Nestler, 2006). Therefore, mechanistic investigations of depression that target hippocampal neurogenesis are critical.

The calcitonin gene-related peptide (CGRP) is a 37-residue amino acid that is contained in the capsaicin-sensitive sensory nerve, is involved in nociceptive transmission, and causes neurogenic vasodilation (Benemei et al., 2009). The CGRP is distributed in the peripheral and central nervous system, including the hypothalamus, central gray matter, amygdala, hippocampus, and dentate gyrus (Skofitsch & Jacobowitz, 1985). Because it has been reported that CGRP is released from the trigeminal terminal in an animal model of migraine, accumulating evidence indicates that CGRP is a major contributor to the onset of migraine (Deen et al., 2017; Iyengar et al., 2019; Naduchamy & Parthasarathy, 2020). However, experimental evidence exists that supports the role of CGRP in depressive-like behavior. Early maternal deprivation leads to reduced levels of CGRP in the hippocampus and occipital cortex (Husum et al., 2002). Our previous study reported that the CGRP mRNA level in the hippocampus significantly decreased after 15-day chronic restraint stress exposure, and CGRP intracerebroventricular administration improved depressive-like behavior in C57BL6J mice (Hashikawa-Hobara et al., 2015). In contrast, in the genetic depressive rat model with maternal separation, CGRP levels were elevated in the frontal cortex, hippocampus, and amygdala (Angelucci et al., 2019). In addition, major depressive disorder patients expressed higher CGRP levels in plasma (Hartman et al., 2006). However, there is still controversy as to how CGRP is associated with depression. Therefore, CGRP-overexpressing mice can be used to evaluate of the influence of CGRP on depressive-like behavior and sensitivity to stress exposure via behavioral tests, which can, in turn, elucidate the underlying molecular mechanisms of CGRP in the hippocampus in response to social defeat stress exposure. Phosphorylated Akt activates various protein kinases and mechanistic targets of rapamycin (mTOR) and is an important downstream effector in the phosphatidylinositol 3-kinase (PI3K) and Akt signal pathways, by which it regulates tumor cell proliferation, angiogenesis, and immunity (Ma et al., 2020; Rho et al., 2020). Furthermore, it has been reported that the ACE inhibitor captopril exhibits an antidepressant effect by activating mTORC1, which is a complex of mTOR, Raptor, mLST8, PRAS40, and DEPTOR, and activates the translation of proteins, which increases the brain-derived neurotrophic factor (BDNF) (Luo et al., 2020). Therefore, the present study assessed whether overexpressed CGRP exerts varying effects on Akt/mTOR signaling in the mice hippocampi under the conditions of social defeat stress. We conducted experiments using previously generated CGRP transgenic (Tg) mice that exhibit high sensitivity to pain and low blood pressure (Mishima et al., 2018).

Materials & methods

Animals

All animal procedures were performed as previously described (Hashikawa-Hobara et al., 2019) and in accordance with the ARRIVE guidelines and approved by the Animal Care and Use Committee of Okayama University of Science. In keeping with these guidelines, efforts were made to minimize the number of animals used and their suffering. All procedures were approved by the institutional ethics committee of the Okayama University of Science (authorization numbers 2016-03, 2017-08, 2018-04, 2019-08, 2020-08, and 1,421, 1,562). Six-week-old C57BL/6J male mice were purchased from Shimizu Experimental Animals (Shizuoka, Japan) and were habituated to a colony for 2 weeks. CGRP transgenic (Tg) mice were generated using standard techniques, as described previously (Mishima et al., 2018). Eight-week-old C57BL/6J mice and Tg mice were used.

All animals were housed in the Animal Research Center of Okayama University of Science at a controlled ambient temperature of 22 °C, with 50 ± 10% relative humidity, and a 12-h light/dark cycle (lights on at 7:00 AM). A total of 120 mice were used. Eleven mice were used for ELISA assays, 12 mice for mRNA or protein analysis, and 97 mice for the behavioral paradigm. The mice showed variation in behavioral test performance; therefore, the group size was set to approximately 10 mice. Biochemical parameters were usually measured in six animals per group. Animals were group-housed, and each home cage (width 235 mm × depth 353 mm × height 160 mm) contained five to six mice with wood flake litter. When breeding transgenic mice, paper nest boxes were used for environmental enrichment.

Social defeat stress (SDS) paradigm

SDS was performed as previously described (Hashikawa et al., 2017). Briefly, each mouse was attacked by a CD1 aggressor mouse for 10 min a day for 15 consecutive days. After being attacked, each pair of mice was separated by an acrylic board with holes to ensure that visual and olfactory stress would continue to be applied. Although the attacked mice suffered wounds, especially to their backs, we withheld analgesic treatment so as to induce stress. If mice had an abnormal appearance (bleeding, immobility) with little chance of recovery, or rapid and sustained weight loss was observed (20% or more in a few days), the experiment was discontinued and the mice were euthanized. After the stress paradigm, behavioral tests were performed. Mice were then deeply anesthetized and blood samples and hippocampi were collected. A total of 1 h before behavioral tests were performed, the mice were moved to the testing laboratory to acclimatize to the environment.

Behavioral assessments

Open field test

Mice were placed in the center of a square open field chamber (30 cm × 40 cm × 20 cm, width × length × height). After a mouse was placed in the center of the open field, all animal behaviors were videotaped using a digital camera (JVC KENWOOD Corp., Tokyo, Japan) for 3 min. The total distance traveled and time (s) spent in the center of the open field were analyzed using Any-Maze behavior tracking software (Muromachi Kikai Co., Ltd., Tokyo, Japan). After the test, mice were returned to their home cage. The floor of the apparatus was cleaned with 70% ethanol after each test.

Social interaction test

Mice were placed in a new area (36 cm × 46 cm × 24 cm, width × length × height; white plastic open field) with a cylindrical metal wire cage (9 cm diameter, 18.5 cm height) at one end in the absence of another mouse, and the mice were allowed to explore freely for 2.5 min. After exploring, the mice were returned to their home cage and allowed to rest. Next, the mice were placed in the same open field chamber, but with an unfamiliar target male ICR mouse enclosed in the cylindrical metal wire cage. Mice were allowed to explore freely for 2.5 min. All behaviors were recorded by digital camera. The time that each mouse spent in the interaction zone was analyzed by Any-Maze behavior tracking software.

Sucrose preference test

The mice were habituated to drinking water from two bottles, one containing tap water and the other 2% (w/v) sucrose in tap water, for 2 days before the end of stress exposure. To encourage the mice to discover the novel sucrose water, the sucrose bottle was initially placed in the usual position for the water bottle, on the right-hand side of the cage, and the water bottle was placed on the left. The two bottles were then switched over on the 2nd day. In the sucrose preference test, two pre-weighed bottles, one containing tap water and the other containing sucrose solution, were presented to each animal for 4 h (10:00–16:00). The position of the water and sucrose bottles was switched every 2 h. The sucrose test was conducted for 2 consecutive days, and the average value was taken as sucrose preference. During this period, each individual mouse was kept in a single cage. Sucrose preference was determined as the percentage of 2% sucrose volume consumed over the total fluid intake volume.

Analysis of the CGRP levels

Mice were anesthetized with a mixture of three anesthetic agents administered intraperitoneally: medetomidine hydrochloride (Domitol, Meiji Seika Pharma Co., Ltd., Tokyo, Japan, 0.3 mg/kg), midazolam (Dormicum, Astellas Pharma Inc., Tokyo, Japan, 4.0 mg/kg), and butorphanol (Vetorphale, Meiji Seika Pharma Co., Ltd., Tokyo, Japan, 5.0 mg/kg) (Kawai et al., 2011). Blood samples were then collected from the aorta. Serum or hippocampus samples were collected from wild-type or Tg mice, respectively. The CGRP levels were measured using a sandwich enzyme-linked immunosorbent assay (ELISA) kit (Bertin Technologies, Montigny-le-Bretonneux, France).

Quantitative analysis by real-time PCR

The animals were killed by an overdose of pentobarbital-Na (100 mg/kg). Total RNA extraction and real-time PCR were performed as previously described (Hashikawa-Hobara et al., 2021). After extraction, RNA samples were reverse transcribed, real-time PCR was performed using Power SYBR Green PCR Master Mix (Life Technologies Japan Ltd., Tokyo, Japan), and samples were analyzed with the Eco Real-Time PCR System (Illumina Inc., Tokyo, Japan).

Information about the primers, which were designed by the authors, is shown in Table 1. The threshold cycle values for the target (Cgrp, Bdnf, and cFos) and internal control (Actin) genes were determined. Data were collected as previously described (Hashikawa-Hobara et al., 2021). Specifically the fold change of each gene was normalized to Actin and was calculated for each sample relative to the expression in the control samples.

Table 1 Oligonucleotide sequences for real-time PCR amplification.

	Forward	Reverse	
Cgrp	CACTGGTGCAGGACTATATGCAG	GTGTTGCAGGATCTCTTCTGAGC	
Bdnf	TGGCTGACACTTTTGAGCACGTC	GCTCCAAAGGCACTTGACTGCTGA	
cFos	GGGACAGCCTTTCCTACTACCAT	GTTGGCACTAGAGACGGACAGAT	
Actin	GGTCAGAAGGACTCCTATGTG	GGTGTGGTGCCAGATCTTCTC	

Western blotting

Western blot analyses were performed as previously described (Hashikawa-Hobara et al., 2021). Briefly, mice hippocampus samples were separated by SDS–polyacrylamide gel electrophoresis and then transferred onto a polyvinylidene difluoride membrane (HybondP; GE Healthcare Ltd., Tokyo, Japan). The membrane was incubated with the following primary antibodies: rabbit polyclonal anti-BDNF (1:5,000, Abcam plc, Cambridge, UK.), rabbit anti phosphor-Akt (Ser473) antibody (1:5,000, Cell Signaling Technology, Tokoyo, Japan), rabbit anti Akt antibody (1:5,000, Cell Signaling Technology, Danvers, Massachusetts, USA), rabbit polyclonal anti-mTOR (phosphor Ser2448) (1:5,000, GeneTex, Inc. International Corporation; Funakoshi Co., Ltd., Tokyo, Japan), rabbit polyclonal anti m-TOR (1:5,000, GeneTex, Irvine, CA, USA), rabbit anti phosphor-p70 S6 kinase (Thr 389/412) antibody (1:5,000, Novus Biologicals; Funakoshi Co., Ltd., Tokyo, Japan), and mouse anti p70 S6 kinase antibody (1:5,000, Santa Cruz Biotechnology. Inc., Dallas, TX, USA). After washing with Tris-buffered saline containing 0.1% (v/v) Tween 20, the membranes were incubated with horseradish peroxidase-conjugated secondary antibody (1:20,000) for 1 h at room temperature. The antibody-reactive bands were visualized using a chemiluminescent substrate kit (Immuno star LD, FUJIFILM Wako Chemicals, Tokyo, Japan).

Statistical analysis

GraphPad Prism 9 software (GraphPad Software Inc., San Diego, CA, USA) was used for all statistical analyses. Data analysis of mice was performed blind to group assignment. We used the Grubbs test to remove outliers from groups. All data are expressed as the mean ± standard error of the mean (SEM). Comparisons between two values were analyzed using Welch’s t test. Two-way analysis of variance (ANOVA) was performed when comparing four values. If there was a significant difference in the interaction between groups (stress and CGRP Tg), Tukey’s post-hoc test was used to compare all groups. A p-value < 0.05 was considered significant. In calculating the size effect, Cohen’s d formula was used and 95% confidence interval (CI) values were calculated for the difference between the average values of wild-type and Tg mice.

Results

Characterization of CGRP-overexpressing transgenic mice

We first determined the level of CGRP in CGRP-overexpressing transgenic (Tg) mice. The serum and hippocampal CGRP levels were significantly increased compared with those of wild-type mice (Fig. 1A, serum; p < 0.0001, d = 30.06, 95%CI [1555–1747], hippocampus protein; p = 0.0014, d = 4.09, 95%CI [16.48–40.88], wild-type (n = 6), Tg (n = 5), hippocampus mRNA; p = 0.0045, d = 2.38, 95%CI [425.3–1832], wild-type (n = 6), Tg (n = 6)). Using C57BL6J mice, we previously reported that CGRP intracerebroventricular administration improves depressive-like behavior while increasing the nerve growth factor and c-fos expression (Hashikawa-Hobara et al., 2015). However, whether conditional CGRP overexpression affects the level of neurotrophic factor in mice hippocampi has not been studied. mRNA transcripts for BDNF and c-fos were significantly increased in the hippocampus compared with those of age-matched wild-type mice (Fig. 1B, Bdnf; p = 0.0018, d = 2.28, 95%CI [34.08–126.6], wild-type (n = 6), Tg (n = 6) or Cfos; p = 0.0013, d = 3.38, 95%CI [189.5–530.1], wild-type (n = 5), Tg (n = 5)). Next, we examined the open field test, which is a comprehensive mouse locomotor test. We recorded and analyzed the travel distance (Fig. 1C). Interestingly, locomotor activity of the Tg mice significantly decreased when compared with that of the wild-type (Fig. 1D, p < 0.0001, d = 3.52, 95%CI [−8.068 to −4.595], wild-type (n = 10), Tg (n = 9)). The locomotor activity of one Tg mouse was 14.28, which was an outlier according to the Grubbs test; therefore, it was excluded. We also performed a social interaction test, which assessed whether Tg mice exhibit normal behavior around an unfamiliar mouse. We found that the time spent approaching the aggressor mouse (ICR mouse) was similar to that of the wild-type mice, which indicated that there were no deficits in social interaction (Fig. 1E, p = 0.2936, d = 0.26, 95%CI [−1.048 to 0.614], wild-type (n = 10), Tg (n = 10)).

Figure 1 Characterization of CGRP Tg mice.

(A) CGRP expression was assessed in the serum, hippocampus, and mRNAs using ELISA or qPCR. (B) Bdnf or cFosmRNA expression in Tg mice compared with wild-type mice. (C) The representative activity in wild-type mice and CGRP Tg mice in the open field test. (D) Locomotion; the total distance traveled. (E) Social interaction rate by measuring time in chambers with social target vs. without social target. Each bar indicates the mean ± SEM. *p < 0.05, Welch’s ttest. Numbers in parentheses indicate the animal numbers for each group.

CGRP-overexpressing mice display stress sensitivity as wild-type mice

Because the expression of BDNF was significantly elevated in the Tg mice hippocampi, we examined whether Tg mice exhibited stress-resistance behavior by performing a battery of behavioral tests. Tg or wild-type mice underwent repeated social defeats during the 15-day social defeat stress procedure. First, we performed the open field test. Mice were randomly divided into four groups: wild-type non-stress (n = 11), wild-type stress (n = 9), Tg non-stress (n = 10), and Tg stress (n = 7). We recorded and analyzed the travel distance or excessive time spent in the center area (Fig. 2A). For distance traveled (locomotor activity), two-way ANOVA showed statistical differences in stress and CGRP Tg interaction (F(1, 33) = 5.890, p = 0.0209) but not stress exposure (F(1, 33) = 2.383, p = 0.1322) and CGRP Tg (F(1, 33) = 3.703, p = 0.0630). Tukey’s post-hoc analysis revealed a significant differences between non-stress wild-type and stress wild-type (p = 0.0285) or non-stress wild-type and non-stress CGRP Tg mice (p = 0.0112). Thus, stress exposure significantly decreased locomotor activity in Tg mice and wild-type mice (Fig. 2B). For time spent in the center area, two-way ANOVA showed statistical differences for stress (F(1, 33) = 10.16, p = 0.0031) and CGRP Tg (F(1, 33) = 5.395, p = 0.0265), but no interaction was detected between stress and CGRP Tg (F(1, 33) = 0.2927, p = 0.5291). Thus, we found that stress and CGRP Tg both shorten the time spent in the center area of the open field test (Fig. 2C). Next, to assess depression-like behavior in Tg mice, we first performed a social interaction test. Although forced swimming or tail suspension tests are commonly used to analyze rodent depressive-like behavior, we decided not to use those experiments because of the significantly reduced locomotor activity of the Tg mice. Therefore, we opted for the widely-used social interaction test and sucrose preference test (Liu et al., 2018) to assess depressive-like behavior. For social interaction rate, two-way ANOVA showed a statistical difference in stress (F(1, 31) = 31.91, p < 0.0001) but not CGRP Tg (F(1, 31) = 1.474, p = 0.2338) and a stress × CGRP Tg interaction (F(1, 31) = 0.8956, p = 0.3513). Thus, we found that the social interaction rate of mice that were exposed to stress was significantly reduced (Figs. 2D, 2E). Next, we performed the sucrose preference test. Mice were randomly divided into four groups with ten mice in each group. Similarly, analysis of anhedonia behavior, which evaluates sucrose preferences, showed statistical differences in two-way ANOVA in stress (F(1, 36) = 23.03, p < 0.0001), but not CGRP Tg (F(1, 36) = 1.137, p = 0.2935) and a stress × CGRP Tg interaction (F(1, 36) = 0.0014, p = 0.994). These results demonstrate that CGRP Tg mice were sensitive to stress.

Figure 2 Social defeat stress exposure manifested as depression-like behaviors in CGRP Tg mice.

(A) The representative activity in CGRP Tg mice with or without stress exposure. (B) Locomotion, the total distance traveled. (C) The time spent in the center area. (D) The representative activity of CGRP Tg mice in social interaction test with aggressor ICR mouse. (E) The social interaction rates. (F) The percentage of the sucrose intake volume over the total fluid intake volume in CGRP Tg mice with or without stress exposure. *p < 0.05, Tukey’s test. Each bar indicates the mean ± SEM. Numbers in parentheses indicate the animal numbers for each group.

Stress exposure stimuli inhibited the Akt-mTOR-p70S6K signaling pathway in CGRP-overexpressing mice hippocampi

To assess why CGRP overexpression did not rescue the stress response even though BDNF levels were adequately upregulated, we evaluated hippocampal BDNF expression in both Tg and stressed Tg mice. For BDNF expression, two-way ANOVA showed differences for CGRP Tg (F(1, 20) = 16.96, p = 0.0005) but not for stress (F(1, 20) = 2.702, p = 0.1159) or a stress × CGRP Tg interaction (F(1, 20) = 0.1757, p = 0.6795). We found that stress exposure did not change the levels of BDNF (Fig. 3A). Next, we examined the expression of phosphorylated Akt and mTOR, which form a signaling cascade from BDNF. Interestingly, stress significantly decreased phosphor-Ser473-Akt and phosphor-Ser2448 mTOR in mice hippocampi [Fig. 3B, CGRP Tg (F(1, 18) = 23.17, p = 0.0001), stress (F(1, 18) = 5.601, p = 0.0294), and stress × CGRP Tg interaction (F(1, 18) = 1.707, p = 0.2078), Fig. 3C, CGRP Tg (F(1, 18) = 6.501, p = 0.0201), stress (F(1, 18) = 5.479, p = 0.0310), and stress × CGRP Tg interaction (F(1, 18) = 3.321, p = 0.085)]. To elucidate whether phosphorylated mTOR induces the downstream target p70S6K, we examined phosphorylated p70S6K expression in mice hippocampi. p70S6K is recognized as a kinase for phosphorylation of 40S ribosomal protein S6 and induces increased protein synthesis (Pullen & Thomas, 1997). We observed that stress exposure significantly reduced phosphorylated p70S6K expression [Fig. 3D, CGRP Tg (F(1, 19) = 4.272, p = 0.0526), stress (F(1, 19) = 12.38, p = 0.0023), and stress × CGRP Tg interaction (F(1, 19) = 1.432, p = 0.2461)]. These data demonstrated that CGRP-overexpressing mice are sensitive to stress exposure because phosphorylated Akt/mTOR/p70S6K expression was down-regulated and the subsequent signaling cascades are inactivated even though BDNF levels are abundant.

Figure 3 Social defeat stress negatively regulates the BDNF signaling pathway in the CGRP Tg mice hippocampus.

(A) The levels of BDNF protein expression were the same even during stress exposure. (B) Phosphorylated Akt expression was reduced by stress. (C) Phosphorylated mTOR expression was reduced by stress. (D) Phosphorylated P70S6K expression was reduced by stress. Each bar indicates the mean ± SEM. Numbers in parentheses indicate the animal numbers for each group.

Discussion

We investigated the effects of CGRP overexpression on the behavior of mice with stress sensitivity to social defeat stress. The typical physiologic features of these mice are low blood pressure and high thermal reaction sensitivity (Mishima et al., 2018). These features are consistent with what is commonly known as CGRP physiological action, such as potent vasodilation, and are involved in the transmission of nociception (Brain et al., 1985; McCulloch et al., 1986). A remarkable behavior phenotype that was discovered in this study is low locomotor activity, which could be due to the high pain sensitivity of Tg mice as previously reported (Mishima et al., 2018). Because CGRP Tg mice exhibited low locomotor activity, we had to abandon the forced swim and tail suspension test, which are the most common tests of depressive-like behavior (Porsolt, Le Pichon & Jalfre, 1977; Cryan, Mombereau & Vassout, 2005). To evaluate depressive-like behavior, we used the sucrose preference test, which measures anhedonic behavior, and the social interaction test. These tests have been reported in chronic stress-induced depression models (Krishnan et al., 2007) in which antidepressants were demonstrated to improve the social interaction of stressed mice (Berton & Nestler, 2006). Previously, we reported that intracerebroventricular administration of CGRP ameliorates depression-like behavior caused by 15-day restricted stress exposure and exerts antidepressant-like effects by increasing the neurotrophic factor (Hashikawa-Hobara et al., 2015). In the present study we observed higher BDNF levels in Tg mice hippocampi; therefore, we expected CGRP-overexpressing Tg mice to exhibit anti-depressive behavior. Contrary to our expectations, CGRP Tg mice did not show stress resistance, but behaved as wild-type mice. One reason for this could be that the systemic effects of CGRP (transgenic mice) and the local effects of CGRP (brain injection) are different. Our previous study showed that injection of CGRP into the brain significantly increased Ngf mRNA but not Bdnf mRNA levels (Hashikawa-Hobara et al., 2015). Thus, increases in neurotrophic factors in response to systemic and local increases in CGRP may differ.

The CGRP receptor activates adenylate cyclase when CGRP binds to its receptor and activates protein kinase A (PKA) through an increase in cyclic adenosine monophosphate (cAMP) (Benarroch, 2011). Furthermore, it has been reported that PKA promotes phosphorylation of Ser133 cAMP-responsive element binding (CREB) and increases BDNF production (Lonze & Ginty, 2002). Therefore, an increase in the BDNF level in Tg mice could be due to CGRP-activated PKA, as increased phosphorylated CREB promotes BDNF transcription. BDNF conditional overexpression mice were observed to be less susceptible to depression (Cryan & Mombereau, 2004; Govindarajan et al., 2006). In addition, the serum levels of BDNF were significantly lower in depressed patients than in healthy subjects but they recovered in response to antidepressant treatments (Shimizu et al., 2003). However, we found that Tg mice had stress sensitivity and down-regulated the BDNF signaling pathway. The results that were obtained in our data showed that stress exposure still induces high-level BDNF expression, but with BDNF signaling cascade down-regulation or desensitization. In support of this notion, we detected decreases in phosphorylated Akt, mTOR, and p70S6K. However, another possible cause is the BDNF precursor (pro-BDNF). BDNF is synthesized and released in a precursor form (pro-BDNF), which is cleaved into mature BDNF that binds to the receptor, TrkB (Lee et al., 2001). In contrast to the neurotrophic effects of BDNF, pro-BDNF binds to the p75 receptor and causes cell death (Teng et al., 2005). Thus, mature BDNF and pro-BDNF exhibit opposite neurotrophic effects. In the present study, we did not examine the level of pro-BDNF or p75 receptor in CGRP Tg mice. Further studies should be conducted to clarify the effect of CGRP overexpression on the BDNF signaling cascade in the mouse brain. Overall, it is reasonable to conclude that numerous effectors are involved but that activation of the BDNF signaling pathway is essential for antidepressant activity.

To characterize the molecular pathways underlying depressive-like behavior in CGRP Tg mice, we focused on phosphorylated Akt and phosphorylated mTOR. mTOR have is reported to be a mammalian target of rapamycin complex 1 and is related to a variety of biological processes. mTOR belongs to the PI3K-related kinases family and is activated by Akt (Costa-Mattioli & Monteggia, 2013). Recently, ketamine, which is a nonselective glutamate N-methyl-D-aspartic acid (NMDA) receptor antagonist, was reported to exhibit antidepressant effects that were activated in the mTOR pathway (Li et al., 2010). In addition, PI3K inhibitor LY294002, or mTOR inhibitor rapamycin administration blocked the antidepressant effect in mice (Ludka et al., 2016). A more recent report also demonstrated that increased BDNF expression activates the mTORC1 pathway and exhibits antidepressant effects (Luo et al., 2020). Thus, reports of antidepressant effects and the BDNF/Akt/mTOR signaling pathway have been accumulating. Furthermore, ribosomal protein S6 kinase beta-1 (S6K1), which is also known as p70S6 kinase, is an enzyme that is activated by mTOR and is a member of cAMP-dependent kinase A. After its activation, S6K1 phosphorylates various substrates and induces protein synthesis (Ghosh & Kapur, 2017). In the present study, social defeat stress exposure significantly reduced the Akt/mTOR/p70S6K pathway, although CGRP Tg mice sustained more than double the amount of BDNF expression in the hippocampus. Therefore, it is reasonable to assume that restoring the Akt/mTOR/p70S6K pathway could be the cause of improvement in the progression of depression.

Although there are few CGRP constitutively overexpressing mice, other reports have prepared CGRP-sensitized mice, which have the human receptor activity modifying protein 1 (RAMP1) subunit of the CGRP receptor overexpressed in the nervous system (nestin/hRAMP1) (Zhang et al., 2007). Unfortunately, the effects of depressive-like behavior have not been tested extensively because the main experimental test was designed for migraine (Russo et al., 2009). In the present study, we evaluated the effects of global CGRP overexpression on depressive-like behavior and the molecular signaling pathway in the mouse hippocampus.

This study has some limitations. First, although a previous study demonstrated that CGRP injection induces an antidepressant effect with increases in the nerve growth factor level in response to chronic restraint stress (Hashikawa-Hobara et al., 2015), the present study did not confirm other neurotrophic factors, except BDNF. Second, the mechanism by which stress down-regulates the Akt/mTOR signaling pathway is still unclear. Other reports have demonstrated that the levels of Akt are decreased in the occipital cortex of depressed suicide victims (Hsiung et al., 2003). In a rodent experiment, the PI3/Akt inhibitor LY294002 suppressed the antidepressant effects of ketamine (Li et al., 2010). Consistent with previous reports, our results revealed that stress exposure down-regulated Akt and inactivated downstream effectors. However, we did not examine whether stress exposure occurred with TrkB receptor desensitization, which induces the uncoupling of G-proteins from the receptor (Gainetdinov et al., 2004). Therefore, additional experiments should be performed to investigate related activities.

Conclusions

The present findings indicate that as the BDNF level in the hippocampus of CGRP-overexpressing mice increased, the mice exhibited the same social activity behavior as wild-type mice. Sensitivity to social defeat stress exposure was at a normal response level because the Akt/mTOR/p70S6K pathway was down-regulated even though the BDNF level was sustained. The findings of this study indicate that the critical mechanism of the BDNF antidepressant effect may not work unless the downstream signal is activated.

Supplemental Information

Supplemental Information 1 Raw data of Figure 1.

Effect size analysis was performed by Cohen’s d test.

Click here for additional data file.

Supplemental Information 2 Raw data of Fig. 2.

Click here for additional data file.

Supplemental Information 3 Raw data of Fig. 3.

The detection was carried using ImageJ. The values obtained from wild-type mice were set to 100%.

Click here for additional data file.

Supplemental Information 4 ARRIVE 2.0 Checklist.

Click here for additional data file.

Supplemental Information 5 Full length blot of Fig. 3B.

Upper gel is BDNF. Bottom gel is Actin.

Click here for additional data file.

Supplemental Information 6 Full length blot of Fig. 3C.

Upper gel is phosphorylated-Akt. Bottom gel is Akt.

Click here for additional data file.

We thank Edanz for editing a draft of this manuscript.

Additional Information and Declarations

Competing Interests

Author Contributions

Animal Ethics

Data Availability

The authors declare that they have no competing interests.

Narumi Hashikawa-Hobara conceived and designed the experiments, performed the experiments, analyzed the data, prepared figures and/or tables, authored or reviewed drafts of the paper, and approved the final draft.

Ami Otsuka performed the experiments, analyzed the data, prepared figures and/or tables, and approved the final draft.

Chihiro Okujima performed the experiments, prepared figures and/or tables, and approved the final draft.

Naoya Hashikawa performed the experiments, analyzed the data, authored or reviewed drafts of the paper, and approved the final draft.

The following information was supplied relating to ethical approvals (i.e., approving body and any reference numbers):

Institute Okayama University of Science provided full approval for this research (2016-03, 2017-08, 2018-04, 2019-08, 2020-08, 1,421, 1,562).

The following information was supplied regarding data availability:

Raw data is available in the Supplemental Files.

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
