# Peer review of "CGRP overexpression does not alter depression-like behavior in mice"

_PeerJ, doi:10.7717/peerj.11720_

## Round 0.1 · original submission · Major Revisions

Two reviewers evaluated your manuscript and provided combinations of minor and major suggestions that would require attention, after which the manuscript can be reconsidered.

Reviewer 1 ·

Basic reporting

the manuscript in general clear and English needs moderate revision (see my general comments attached)

Experimental design

Experiments are well designed and conducted and methods sufficiently described.
There is only one major issue in the results figure 3, some important control groups are missing (see my general comments attached)

Validity of the findings

the findings looks robust and are well discussed based on literature

Additional comments

The authors describe interesting experiments evaluating the effects of the overexpression of CGRP in a mouse model of stress-induced depression.
Althrough in their model the overexpression of CGRP do not change depression like behavior it highlight how CGRP can affects BDNF downstream pathway in the hypocampus. The rational of the project was clearly presented and results are well discussed based on the current literature.

Major comments:

Figure 3 presents the most important results of the paper, however, control groups are missing to really appreciate the effects of CGRP on AKT/mTOR/P70s6K pathway.
The authors should include the AKT, mTOR and P70s6K western blot data and quantification for wt and wt stress groups to know if by itself the overexpression of CGRP affect AKT/mTOR/P70s6K pathway compare to wt, and how stress affects this signaling pathway by comparing wt stress versus tg stress groups.

In a previous paper the authors showed that icv injection of CGRP reduced depression behavior, but do not find the same results with the tg mice.
It would be interesting to discuss the difference between the local effects of CGRP (icv injection) and the systemic effects of CGRP (transgenic mice) on depression.


Other comments
In the abstract the authors say “which indicates that CGRP Tg mice exhibit stress-resistance”; as stress and depression are linked it is awkward to limit stress-resistance to only the increase of BDNF.

The manuscript should be revised for language.
Line 108-109 authors means “we did not treat them with analgesia to not interfere the exposition to stress”
Line 113 the authors mean “hippocampi were collected after deep anesthesia and euthanasia” as it is more ethical to euthanise the mouse before collecting brain tissue.
Line 308 authors mean, “by increasing the neurotrophic factor BDNF”
Line 319-320 the sentence should be rephrased for ore clarity.

Reviewer 2 ·

Basic reporting

There are several (minor) language errors throughout the manuscript that must receive attention. For instance, in line 80: "This Tg mouse expresses high sensitivity of pain [and] low blood pressure...". Line 108/9: "...we did not treat analgesia them to expose stress." Line 113: "...were collected at the deep anesthesia." Line 198: "...was performed when compared four values were analyzed." These sentences (and others) require improvement. Further, the use of capital and lower case letters in the figures and text are not constant. For instance, CGRP (text) vs cgrp (figures).

The introduction section is well written and gives sufficient context for the purpose of the study. The use and specific characteristics of the animal model used here (i.e. CGRP Tg mouse) can however be briefly elaborated on. The section concludes with reference to the increased pain sensitivity and hypotension-like characteristics of the model. A brief description of its CGRP profile would improve the context of the manuscript for the upcoming results and discussion.

The structure of the article is professional with the figures accurately representing the reported data. All raw data was also shared - thank you.

Experimental design

The research question is relevant to the aims and scope of the journal.

The research question is clearly stated and relevant to the knowledge gap identified in the Introduction section of the manuscript.

The methodology is well described and would enable other groups to replicate the study design. The authors must however please elaborate on the sucrose preference test's method. In this regard, the authors earlier state that animals were group housed, yet the data of the sucrose preference test is individual data points, suggesting that the animals were individually housed for the 4 hours of the sucrose preference test? If this is indeed the case, the authors must please include this in their method section, as single-housing could have a significant effect on behaviour and can then at least be considered by the reader.

Also, were the behavioural tests performed on separate days or as a battery of tests?

Validity of the findings

The major comment for the authors (and editors) to consider is the statistical methods used in the current manuscript. The authors state that two-way ANOVAs were used as well as "one-way ANOVAs... where appropriate." (lines 199/200). From the Results section, it appears as though two-way ANOVA results were followed up with a one-way ANOVA - why is this? The two-way ANOVA results are accurately reported, with both interactions and main effects available. In instances where only an interaction between stress and strain (CGRP Tg) is significant, the authors can determine inter-group differences with appropriate multicomparison (post-hoc) tests, such as Tukey or Bonferoni. In other instances where only the main effects were statistically significant, the authors should interpret these results irrespective of the other factor and not re-evaluate the data with a one-way ANOVA. This is of particular concern, as a one-way ANOVA assumes a single factor to affect the results, which in the case of the manuscript is not the case. In Figure 2C, the authors report that no significant interaction existed, yet compare the WT (non-stress) and Tg (stress) groups. This is statistically incorrect and must be reconsidered.

Secondly, although the authors describe the method used to identify and exclude outliers and also report where data points were excluded, statistical tests to identify such data points (e.g. Grubbs tests) can also be considered.

The authors also report a 95% CI in the Results section - what does this indicate? Is it the CI of the mean difference? Please calrify.

In the current format, the conclusion made from the results could be inaccurate and must therefore be re-evaluated and considered.

Additional comments

The authors investigated a relevant and interesting research question, with the potential to contribute to our current understanding of depression and possibly identify novel treatment targets. However, to improve the drawn conclusions and potential impact, the authors must please reconsider their statistical analysis methods.

---

## Round 0.2 · Minor Revisions

The reviewers were approving of the major issues raised, with only minor revisions to attend to. The second reviewer, however, still has one concern about the statistical approach that needs to be addressed.

Reviewer 1 ·

Basic reporting

na

Experimental design

na

Validity of the findings

na

Additional comments

All my comments have adequately been addressed.
However, I have an additional minor comment to make before the manuscript is suitable for publication.

In the discussion line 304, “Thus, increases in neurotrophic factors in response to systemic and acute increases in CGRP may differ”, do the Authors mean “local” rather than “acute”?

Reviewer 2 ·

Basic reporting

The authors have addressed the concerns raised in the previous review. Thank you for improving the language of the manuscript.

Experimental design

All concerns have been addressed by the authors, improving the manuscript and clarifying where needed.

Validity of the findings

I remain concerned with the statistical methods used. Although improvements have been made (thank you), the use of a one-way ANOVA as follow-up test for two-way ANOVAs is concerning. For instance, the results of Fig 2B (line 255) state: "...two-way ANOVA showed statistical differences in stress and CGRP Tg interaction (F(1,33) = 5.890, p = 0.0209) but not for [main effects]. One-way ANOVA (F(3,33) = 4.662, p = 0.008) detected significant differences..." Why was the two-way ANOVA followed up by this test, specifically in the presence of a significant interaction? An appropriate Tukey (or other) post-hoc test performed here, would be able to identify group differences. Further, because of there being two influential factors (i.e stress and Tg), a one-way ANOVA cannot be performed as this test assumes only one influential factor. Although it may be that the reported results may be unchanged following other analyses, the repeatability of this data set as a whole would still be greatly improved. I therefore suggest the authors to consult statistical services for confirmation.

Additional comments

Thank you for providing an improved version of the manuscript. Because of the concern regarding the statistical analyses, I suggest consulting a bio-statistician / statistical services for confirmation, to validate the discussion and overall conclusion.

---

## Round 0.3 · accepted · Accept

Thanks for addressing these last minor suggestions, the manuscript can be accepted in this final form.